# Dynamic Changes of Bacterial Communities and Microbial Association Networks in Ready-to-Eat Chicken Meat during Storage

**DOI:** 10.3390/foods11223733

**Published:** 2022-11-21

**Authors:** Mengjia Qiu, Xingning Xiao, Yingping Xiao, Jiele Ma, Hua Yang, Han Jiang, Qingli Dong, Wen Wang

**Affiliations:** 1School of Health Science and Engineering, University of Shanghai for Science and Technology, Shanghai 200093, China; 2State Key Laboratory for Managing Biotic and Chemical Threats to the Quality and Safety of Agro-Products, MOA Laboratory of Quality & Safety Risk Assessment for Agro-Products (Hangzhou), Institute of Agro-Product Safety and Nutrition, Zhejiang Academy of Agricultural Sciences, Hangzhou 310021, China; 3Key Laboratory of Specialty Agri-Products Quality and Hazard Controlling Technology of Zhejiang Province, College of Life Sciences, China Jiliang University, Hangzhou 310018, China

**Keywords:** ready-to-eat chicken meat, bacterial community, microbial association networks, high-throughput sequencing, 16S rDNA

## Abstract

Ready-to-eat (RTE) chicken is a popular food in China, but its lack of food safety due to bacterial contamination remains a concern, and the dynamic changes of microbial association networks during storage are not fully understood. This study investigated the impact of storage time and temperature on bacterial compositions and microbial association networks in RTE chicken using 16S rDNA high-throughput sequencing. The results show that the predominant phyla present in all samples were *Proteobacteria* and *Firmicutes*, and the most abundant genera were *Weissella*, *Pseudomonas* and *Proteus.* Increased storage time and temperature decreased the richness and diversity of the microorganisms of the bacterial communities. Higher storage temperatures impacted the bacterial community composition more significantly. Microbial interaction analyses showed 22 positive and 6 negative interactions at 4 °C, 30 positive and 12 negative interactions at 8 °C and 44 positive and 45 negative interactions at 22 °C, indicating an increase in the complexity of interaction networks with an increase in the storage temperature. *Enterobacter* dominated the interactions during storage at 4 and 22 °C, and *Pseudomonas* did so at 22 °C. Moreover, interactions between pathogenic and/or spoilage bacteria, such as those between *Pseudomonas fragi* and *Weissella viridescens*, *Enterobacter unclassified* and *Proteus unclassified*, or those between *Enterobacteriaceae unclassified* and *W.viridescens*, were observed. This study provides insight into the process involved in RTE meat spoilage and can aid in improving the quality and safety of RTE meat products to reduce outbreaks of foodborne illness.

## 1. Introduction

Chicken consumption has increased rapidly in China, and the annual consumption amount has reached 15.46 million tons, representing an 11.21% increase from 2019 [1,2]. Ready-to-eat (RTE) meat products are pre-cooked meat that can be consumed either directly or with minimal preparation, and RTE chicken has become popular not only due to consumers’ convenience, but also because of the high levels of protein, iron, vitamins, selenium and niacin, as well as reduced fat and cholesterol [3]. RTE meat products are susceptible to microbial contamination, including spoilage and pathogenic microorganisms captured during processing, storage and transportation [4]. Although the viable counts of microorganisms during the storage of RTE chicken products have been extensively reviewed, few studies have focused on the dynamic changes in the microbial associations that accompany spoilage-associated bacteria and potential pathogens under different storage times and temperatures in RTE chicken [5,6,7,8].

Traditional studies on bacterial diversity in foods have relied on cultivation methods, whereas only about 1% of microorganisms in the natural environment can be cultivated through pure culture methods [9]. Culture-independent technologies that involve molecular biology approaches, such as gradient gel electrophoresis (DGGE) and polymerase chain reaction (PCR), have been applied to the evaluation of microbial diversity in a variety of environments, including foods, avoiding the limitations of traditional microbial analyses [10,11,12,13]. These measurement methods have been expanded using high-throughput sequencing (HTS) to analyze microbial community structures [9]. In addition, HTS techniques have indicated that microbial interactions during food storage are critical in shaping the microbiota. Analyses of microbial association networks can function to uncover previously unexplored interactions between microbial taxa and identify species that act as hubs, i.e., they have many interactions with other species [14]. Numerous bioinformatics methods are in use that can reveal microbial association networks, including Co-occurrence Network Inference (CoNet) [15], Sparse Correlations for Compositional data (SparCC) [16] and Sparse Inverse Covariance Estimation for Ecological Association Inference (SPIEC-EASI), which has been recently been utilized for the American Gut Project [17]. SparCC is considered valuable since it corrects for spurious correlations to identify true associations missed by Pearson correlations and is a robust method to determine compositional effects that are influenced by the correlation diversity and sparsity in human microbiome data sets [18]. In addition, SparCC could detect the largest number of significant associations that were either positive (co-present) or negative (mutually excluded), while SPIEC-EASI generated the lowest number of associations [19]. The relative strengths and weaknesses between CoNet and SparCC have been assessed using synthetic data, and SparCC surpassed CoNet in terms of accuracy, sensitivity and precision [20]. However, the application of microbial association networks to the study of food microbial communities is poorly exploited. Exploring the correlation between beneficial and spoilage-associated/potentially pathogenic bacteria will provide useful information for improvement of food quality and safety, as well as new approaches for identifying hub species in food microbial communities [19]. In the current study, we applied HTS to provide a general background of bacterial communities in RTE chicken storage at different temperatures and times. Meanwhile, we inferred microbial association networks using SparCC in order to investigate the structure and properties of a variety of bacterial association networks in RTE chicken, and to further explore the potential microbial risks during storage.

## 2. Materials and Methods

### 2.1. Sample Collection and Processing

In this study, the RTE chicken samples were purchased from a local supermarket, where the whole chicken was cooked by heating in boiling water for 15–20 min with ginger, scallion and garlic, then sliced and packaged in a polypropylene (PP) plastic box, stored under normal atmospheric conditions in the counter and sold as RTE chicken. Samples were transported to the laboratory within 1 h in an insulated box containing an ice pack and then stored separately at refrigerated temperatures representing optimal (4 °C) and suboptimal (8 °C) conditions, as well as at room temperature (22 °C). Five samples were analyzed for bacterial communities in RTE chicken on Day 0 (Group O). The others were analyzed on the 1st, 2nd, 3rd, 4th and 5th days at 4, 8 and 22 °C, which were named L1–L5 (Group L), M1–M5 (Group M) and H1–H5 (Group H), referring to the 4, 8 and 22 °C storage on Days 1–5, respectively. For each analysis, eight parallel samples were used.

### 2.2. DNA Extraction

The samples were individually added to sterile stomacher bags containing 225 mL of buffered peptone water (Beckton Dickinson, Franklin Lakes, NJ, USA) and kneaded gently for 2 min. Then, 40 mL of the mixture was centrifuged at 4000× *g* for 10 min and the supernatant was again centrifuged at 12,000× *g* for 5 min. The DNA from different samples was extracted using the TIANamp Stool DNA Kit (Tiangen Biotech, Beijing, China) according to the manufacturer’s instructions. The DNA concentration was measured using a Nanodrop One spectrophotometer (Thermo Fisher, Pittsburg, PA, USA), and the DNA quality was confirmed using 1% agarose gel electrophoresis. The total DNA was stored at −20 °C until PCR analysis.

### 2.3. PCR Amplification and Sequencing

The V3–V4 region of the bacterial small-subunit 16S rRNA gene was amplified with slightly modified versions of primers 341F (5′-CCTACGGGNGGCWGCAG-3′) and 805R (5′-GACTACHVGGGTATCTAATCC-3′) [21]. PCR amplification was performed in a total volume of 25 μL of the reaction mixture, containing 25 ng of template DNA, 12.5 μL of PCR Premix, 2.5 μL of each primer and PCR-grade water to adjust the volume. The PCR conditions to amplify 16 S rDNA consisted of an initial denaturation at 98 °C for 30 s, followed by 32 cycles of 98 °C for 10 s, 54 °C for 30 s, 72 °C for 45 s and a final extension at 72 °C for 10 min. The PCR amplicons were confirmed using 2% agarose gel electrophoresis, gel-purified using an AMPure XT kit (Beckman Coulter Genomics, Danvers, MA, USA) and quantified using a Qubit instrument (Invitrogen, Carlsbad, CA, USA). Amplicon pools were prepared for sequencing, and the size and quantity of the amplicon library were assessed with an Agilent 2100 Bioanalyzer (Agilent, Santa Clara, CA, USA) and the Library Quantification Kit for Illumina (Kapa Biosciences, Woburn, MA, USA), respectively. 

### 2.4. Sequence Data Analysis

The samples were sequenced on an Illumina NovaSeq platform using a 2 × 250 cycle kit provided by LC-Biotech, Hangzhou, China, according to the manufacturer’s recommendations. Paired-end reads were assigned to samples based on their unique barcode, truncated by cutting off the barcode and primer sequence and then merged using FLASH (Version 1.2.8, Macromedia, San Francisco, CA, USA) to an average length of 426bp [22]. The sequencing quality was assessed with fastqc (http://www.bioinformatics.babraham.ac.uk/projects/fastqc/, accessed on 20 September 2020). Quality filtering on the raw reads was performed under specific filtering conditions to obtain high-quality clean tags using fqtrim (Version 0.94, Johns Hopkins University, Baltimore, MD, USA). Chimeric sequences were filtered using Vsearch software (Version 2.3.4, https://github.com/torognes/vsearch, accessed on 20 September 2020). The Operational Taxonomic Units (OTU) table, which selected reads with similarities of 100%, was obtained after dereplication using DADA2 [23]. The sequence data obtained in this study were deposited in NCBI BioProject PRJNA744008.

The alpha and beta diversity were calculated using random normalization to the same sequences. Alpha diversity was applied in analyzing the complexity of species diversity for a sample through five indices, including Chao1, Observed species, Goods coverage, Shannon and Simpson, calculated using QIIME2 (https://qiime2.org/, accessed on 20 September 2020) [24]. Beta diversity, which refers to species differences between different environmental communities, included principal coordinates analysis (PCoA) and clustering analysis (UPGMA), calculated with QIIME2. Graphs were drawn using the R package (Version 3.4.4, http://www.r-project.org/, accessed on 20 September 2020) [25]. BLAST (https://blast.ncbi.nlm.nih.gov/Blast.cgi, accessed on 20 September 2020) was used for sequence alignments, and the feature sequences were annotated with the SILVA database (http://www.arb-silva.de/, accessed on 20 September 2020) for each representative sequence. Redundancy analysis (RDA), heatmaps, cluster and correlation analyses were performed using OmicStudio (https://www.omicstudio.cn/tool, accessed on 5 May 2021). Genera with relative abundances above 0.01% were selected to construct the co-occurrence network. To investigate co-associations among bacterial taxa, we used the network inference tool SparCC, which is based on an iterative approximation approach and uses log-ratio transformed data to infer the correlations between components. SparCC’s correlation was estimated using the psych package in R. Robust correlations were defined as those with the Pearson correlation coefficient threshold values of >0.2 and *p* < 0.05 [26]. The network analysis was conducted using the igraph (1.2.6) package in R (http://igraph.org/, accessed on 5 May 2021 ). The pathogens were classified as previously reported and included the list of human pathogens and the German Technical Rules for Biological Agents [27,28]. 

### 2.5. Statistical Analysis

The SPSS 20.0 software package (SPSS, Chicago, IL, USA) was used for statistical analysis. The data are shown as the mean ± standard deviation (S.D.) for every group. The Kruskal-Wallis test was used for diversity differential analysis. ANOSIM was performed based on the Bray-Curtis dissimilarity distance matrices to identify differences in the microbial communities among different groups. The significance level was set at *p* <0.05.

## 3. Results

### 3.1. Characteristics of OTUs in RTE Chicken Samples during Storage

The sequencing resulted in 64,947 average valid sequence reads in each sample, fewer than the 77,645 raw sequences. Singleton reads were not considered for subsequent analyses. A total of 7,272,857 high-quality effective sequences with a mean of 58,182 sequences per sample clustered into 5122 OTUs. The Good’s coverage estimator of the completeness of sampling was at least 0.999, indicating that the sequencing reads covered almost all the bacterial populations present in the samples [29,30] (Table 1). The minimum sample sequence number of 38,427 was sufficient to reflect the diversity of the bacterial species in each sample, and the sequencing amount met the requirements of subsequent bioinformatics analysis. 

The alpha diversity indices (Observed OTU, Shannon, Simpson and Chao1) were used to measure and compare the microbial diversities in the samples during storage. As shown in Table 1, with the increasing storage time and temperature, the Observed OTU and Chao1 indices generally decreased, indicating that the richness and diversities of microorganisms in the RTE chicken were decreased. As shown in Figure 1A, 2617, 2302 and 2264 OTUs were obtained from the samples stored at 4, 8 and 22 °C, respectively, among which 686 OTUs were observed to be common in all samples. In addition, 404, 1689, 1818, 1818, 1489 and 1554 were obtained from samples collected on Days 0–5, among which 98 OTUs were observed to be common in all samples. Interestingly, among the 98 common OTUs, *Serratia proteamaculans* (OTU ID 34588d48b1f866ee34ee3225e78009e8), which could be spoilage bacteria and potentially pathogenic bacteria, was present in all samples (mean of 2.41%) and deserves further attention.

### 3.2. Taxonomic Composition of Bacterial Community

The 16S rDNA gene sequencing showed that the microbial communities of all samples covered 18 phyla, 42 classes, 86 orders, 165 families, 387 genera and 1374 species. As shown in Figure 2A, *Firmicutes* was the most dominant phylum (mean of 88.66%) on RTE chicken samples collected from the supermarket (Group O), followed by *Proteobacteria* (mean of 11.91%) at the phylum level. When stored at 4 °C (Group L) and 8 °C (Group M), the relative abundance of *Proteobacteria* gradually increased with the extension of storage time, achieving maximum abundances of 81.74% and 97.45% on Day 5, respectively. When stored at 22 °C (Group H), the relative abundance of *Proteobacteria* rapidly increased to 90.38% on the first day, but decreased to 83.23% on Day 5. During storage, the relative abundance of *Bacteroidetes* gradually increased to 20.19%, becoming another dominant phylum.

At the genus level, the bacterial communities showed a dramatic increase in complexity with increases in storage temperature and time (Figure 2B). On Day 0, *Weissella*(87.02%) was the most predominant. At 4 °C, the relative abundance of *Weissella* gradually decreased to 16.62% on Day 5, and the relative abundance of *Pseudomonas* increased to 50.70%, becoming the dominant genus. At 8 °C, the relative abundance of *Weissella* decreased to less than 1% on Day 5, and *Serratia*, *Pseudomonas* and *Acinetobacter* increased to 24.81%, 17.02% and 13.48%, respectively, becoming the dominant genera. At 22 °C, the relative abundance of *Weissella* decreased rapidly to 1.64% on Day 1, and *Proteus* and *Myroides* grew to be the dominant genera, with relative abundances of 46.19% and 15.85% on Day 5, respectively.

The bacterial species’ richness for samples stored at 4 and 8 °C was generally higher than that at 22 °C (Figure 2C). *W. viridescens*, the dominant species on Day 0, with an average relative abundance of 58.27%, decreased to 14.53% on Day 5, and *P. fragi* (17.71%) became the most predominant species. At 8 °C, the relative abundance of *W. viridescens* and *Weissella minor* gradually decreased, and *E. unclassified* and *S. proteamaculans* increased to 14.12% and 15.31%, respectively, on Day 5. The three most abundant species, including *P. unclassified*, *Myroides unclassified* and *Providencia unclassified*, showed increasing trends at 22 °C, reaching 42.06%, 12.54% and 6.94%, respectively, on Day 5.

### 3.3. Microbiota on RTE Chicken Meat Varies over Time during Storage at Different Temperatures 

Bacterial communities were further compared using PCoA, which showed that an examination of the score plots in the area defined by the first two principal components accounted for 66.92 and 15.1% of the total variance, and significant differences were observed among all groups for PCoA (*p* = 0.001). The samples stored at 4 and 8 °C partially overlapped, indicating a similarity in their bacterial compositions. The analysis showed discrimination between the samples at 22 °C and those at the other two temperatures, with no overlap, indicating that the higher storage temperature had a significant impact on the bacterial community composition (Figure 3A,B).

In order to reveal the effects of storage time and temperature on the composition and similarity of bacterial communities, sample clustering was performed using Bray-Curtis distances. In general, the products were clearly clustered according to storage time and temperature. As shown in Figure 3C, the samples from the first and second days were distant from those of the last three days, and samples stored at 4 and 8 °C were clustered together, while those at 22 °C were clearly separated (Figure 3D and Appendix A). The RDA analysis had concordant results, showing that storage temperature and time had strong and positive effects on the distribution of the bacterial community of the RTE chicken meat samples (Figure 3E).

We used heat maps to visually define the groups that contributed to the differences in the bacterial community composition. The samples at 22 °C displayed relative abundances of *Klebsiella*, *Enterobacter*, *Citrobacter*, *Pluralibacter*, *Morganella*, *Arcobacter*, *Myroides*, *Proteus*, *Providencia* and *Stenotrophomonas,* which exceeded the levels for the samples at 4 and 8 °C. Meanwhile, the samples stored at 4 and 8 °C displayed similar heat map profiles. As for the groups with different storage times, there was a significant difference between Day 0 and the other days. The relative abundances of *Pseudomonas, Psychrobacter*, *Acinetobacter*, *Morganella*, *Shewanella*, *Erwinia*, etc. on Day 0 were significantly lower than those on the other days (Appendix A).

### 3.4. Potentially Pathogenic and Spoilage-Associated Species

Bacterial pathogens, including *P. unclassified*, *M. unclassified*, *E. unclassified*, *Acinetobacter* sp., *Psychrobacter* sp., *S. proteamaculans*, *P. unclassified*, *Pantoea agglomerans* and *Pseudomonas* sp., were identified in our samples, and are associated with urinary tract infections, bacteremia, pneumonia, diarrhea, septicemia and meningitis (Table 2). As shown in Figure 4A, the total relative abundance of potentially pathogenic species increased over time at all storage temperatures, and higher total relative abundances were observed at 22 °C. *Enterobacteriaceae*, *Acinetobacter* sp. and *Psychrobacter* sp. were dominant pathogenic species at 4 and 8 °C, while *P. unclassified* and *M. unclassified* became dominant at 22 °C. Notably, familiar foodborne pathogen, such as *Salmonella*, *Escherichia coli* and *Listeria*, were almost undetectable (accounting for less than 0.1% of the total abundance), indicating that further validation is required by using targeted qPCR or a traditional culture method.

Spoilage-associated species, including *W. viridescens*, *Pseudomonas* sp., *Enterobacteriaceae*, *Acinetobacter* sp., *Psychrobacter*sp. and *S. proteamaculans,* were also identified, which are responsible for the development of off-odors, volatile spoilage compounds, putrescine and cadaverine, thereby making it unacceptable for human consumption (Table 2). As shown in Figure 4B, the total relative abundances of spoilage-associated species kept relatively stable when stored at 4 and 8 °C, while they decreased significantly at 22 °C. *W. viridescens* was the most dominant species at 4 °C, while it decreased rapidly at 8 °C. *E. unclassified*, *S. proteamaculans* and *Psychrobacter* sp. gradually increased during storage at 8 °C, becoming the dominant spoilage-associated species. At 22 °C, the total relative abundances of spoilage-associated species were much lower than those at 4 and 8 °C. *W. viridescens* was almost unrecognized, and *Acinetobacter* sp. and *Psychrobacter* sp. became the dominant spoilage bacteria.

### 3.5. Microbial Association Networks during Storage

In microbial association networks, positive interactions might represent cooperation or complementation among species, while negative interactions might signify competition, predation or amensalism [50,51]. In this study, networks at the genus and species level were constructed (Figure 5). At the genus level, 22 positive and 6 negative interactions at 4 °C, 30 positive and 12 negative interactions at 8 °C and 44 positive and 45 negative interactions at 22 °C were observed (Figure 5A–C). At the species level, 49 positive and 36 negative interactions, 80 positive and 39 negative interactions and 73 positive and 69 negative interactions were identified at 4, 8 and 22 °C, respectively (Figure 5D,E). In general, potential interactions became more complex as the storage temperature increased.

As shown in Figure 5 and Table 3, *Enterobacter* was the core genus in the microbial association networks at both 4 and 22 °C, and *Pseudomonas* was observed as the core genus at 8 °C. When stored at 4 and 8 °C, there was a strong negative correlation between *Pseudomonas* and *Weissella* (r = −0.702 and r =−0.678, respectively). At 22 °C, *Enterobacter* was negatively correlated with *Proteus* (r = −0.578). The results are consistent with the changes in the relative abundances of *Pseudomonas*, *Weissella* and *Enterobacter* during storage in Figure 2B.

At the species level, *P. fragi*, *W. viridescens* and *P. unclassified* were the main bacteria that constituted the microbial association networks at 4, 8 and 22 °C, respectively (Appendix A). The highest positive correlation was observed between *W. minor* and *W. viridescens* among the networks at 4 °C (r = 0.653) and 8 °C (r = 0.722). Consistent with the results in Figure 2C, *P. fragi* was negatively correlated with *W. minor* at 4 °C (r = −0.622), and *S. proteamaculans* was negatively correlated with *W. viridescens* at 8 °C (r = −0.526). At 22 °C, *P. unclassified* and *P. unclassified* had a positive correlation (r = 483), while *P. unclassified* and *Klebsiella aerogenes* had a negative correlation (r = −0.578).

## 4. Discussion

Initial indigenous microbiota are derived from raw materials, ingredients and additives, as well as from the external environment, including roasting, packaging, storage and operators, accounting for high microbial diversity at the initial storage times [52,53]. Dynamic changes in bacterial communities significantly affected the quality and safety of RTE meat. The core microbiota in our samples were dominated by *Proteobacteria* and *Firmicutes*, which is in agreement with the previously published study [54]. These phyla were not significantly different in the 4 and 8 °C samples, indicating that *Proteobacteria* and *Firmicutes* might be the primary phyla responsible for meat spoilage at cold storage temperatures [55]. *Pseudomonas,* which belongs to *Proteobacteria,* is often sourced from the environment, especially the meat processing plant. Some *Pseudomonas* species are recognized as spoilage-associated species, since they have the ability to produce pectinolytic enzymes that cause meat spoilage [56]. In this study, *Pseudomonas* generally increased in relative abundance during storage at 4 and 8 °C, indicating this genus may be a driving force for the spoilage of RTE chicken at low temperatures. Furthermore, in the early stages of low temperature storage, the relative abundance of *Weissella* was elevated and gradually decreased with time, indicating that this bacterium may be at a competitive disadvantage compared to the dominant genera that drive spoilage of RTE chicken meat [57]. *Proteus* was the dominant genera at 22 °C, which differed from those at 4 and 8 °C. However, *Proteus* sp. is sensitive to pasteurization and common disinfectants, weakly resistant to heat and unlikely to bear viable *Proteus* cells in meat if correct cooking steps have been taken. Thus, *Proteus* in RTE chicken can originate from packing, transportation or storage [58].

Interactions among microorganisms are important in any food in which a mixed flora develops during storage [59] and microbial interaction networks can be used to predict hub species and potential species interactions in these instances [60]. Genus-level analyses are associated with the highest connectivity, and genera are typically considered the keystone taxa in a network [61]. In our study, *Pseudomonas* dominated in interactions at 4 and 8 °C, while *Proteus* dominated at 22 °C (Figure 5). *Pseudomonas* also displayed negative effects on *Enterobacter*, *Weissella*, *Klebsiella* and *Aeromonas.* It has been shown that *Pseudomonas* could inhibit some spoilage-associated and pathogenic bacteria via siderophore-mediated competition for iron or competition for specific nutrients [62,63]. In addition, *Pseudomonas* inhibited the growth of foodborne pathogens via cell-contact-dependent competition established in the food matrix, which can be used as a non-probiotic antagonistic bacterium to reduce the risk of foodborne pathogens [64]. We found that *Proteus* had 13 negative interactions with *Klebsiella*, *Enterobacter*, *Pantoea*, *Serratia*, etc. *Klebsiella* and *Enterobacter* belong to potentially pathogenic bacteria. *Serratia* and *Pantoea* are primary spoilage bacteria in meat and meat products during aerobic storage [65,66]. Therefore, *Proteus* can be used as a basis for biomarker research to find harmful bacteria [67]. *P. unclassified* was negatively correlated with 23 species and consistent with these observations. In addition, in low-temperature conditions, *W.viridescens* had negative interactions with *E. unclassified*, *Pseudomonas plecoglossicida*, *Serratia proteamaculans*, *Psychrobacter* sp. and other potential pathogens. Although *W.viridescens* is often regarded as the dominant spoilage bacterium, it displays a wide range of antimicrobial activities against potential pathogens through the production of antimicrobial compounds, including *Bacillus cereus*, *Clostridium botulinum*, *Escherichia coli* and *Listeria monocytogenes* [9,68,69]. This could be advantageous in extending the microbiological safety of these products. Microbial interactions are not only due to the competition fornutrients, but also to cell-to-cell communication (quorum sensing) and metabolism (creating an unfavorable environment), which may affect microbial behavior [70]. Using microbial interactions to inhibit the growth of harmful bacteria can be a novel strategy to improve food quality and safety.

Taxonomic identification is a problem to be solved when studying food microbiota using HTS. The classical V3–V4 region of the16S rDNA (usually family- or genus-level) lacks accurate taxonomic assignment at the intra-species level. As shown in Figure 4, *Enterobacteriaceae* was ubiquitously present in all samples. Members of *Enterobacteriaceae* are widely distributed and include a wide range of important enteric foodborne pathogens, such as *Shigella* sp., *Salmonella* sp. and *Escherichia coli*, which represent a strong threat to public health and food safety [71,72]. In the current study, we only identified *Enterobacter cancerogenus* (mean of 0.76%) among the top 30 species in relative abundance, and others were included in *Enterobacteriaceae unclassified* (mean of 3.12%) and *Enterobacter unclassified* (mean of 0.72%). This is due to the close sequence identities of *Enterobacteriaceae*, which makes the species-level resolution of *Enterobacteriaceae* challenging with 16S rRNA amplicon sequencing [73]. Therefore, more studies with whole-genome sequencing, shotgun sequencing of metagenomes and other molecular biology tools are needed to provide greater taxonomic and functional information at the species level [74]. Additionally, *Serratia*, *Enterobacter*, *Klebsiella*, *Rahnella*, etc. could be found ubiquitously in samples, and they were reported to contribute significantly to the spoilage flora on meat and meat products. *Enterobacter* and *Klebsiella* decreased and almost disappeared during storage, which suggested that they did not play a crucial role in meat spoilage [53].

Further investigation of bacterial interactions during storage could provide new strategies for reducing and inhibiting the growth of potential pathogens and spoilage-associated bacteria. Several models for this process have been developed, including the Jameson-effect [75] and Lotka-Volterra [76] models, which predict interactions between potential pathogens and spoilage-associated bacteria. Nevertheless, at present, the research on food bacterial communities lacks large time-series studies, and the nature of data obtained from HTS for foods (which are relative abundances rather than absolute quantitative values) is not completely adequate for evaluating interactions and developing an interaction model under different storage conditions in microbial communities. Therefore, further work should be performed to obtain quantitative changes in bacteria during the storage of RTE meat products and to develop more realistic interaction predictive models.

## 5. Conclusions

This study revealed that different storage times and temperatures affected the composition of bacterial communities and the microbial association network. Higher storage temperatures impacted the bacterial community composition and microbial association network more significantly. The relative abundance of potentially pathogenic bacteria increased during storage, representing a significant threat to human health. The core bacteria which dominated the interactions during storage, such as *Pseudomonas* and *Proteus,* can be considered the critical control bacteria for the microbiological quality of RTE chicken. The interactions detected in this study primarily reflect co-occurrence and mutual exclusion patterns at different storage temperatures, although some cases suggested true positive (commensalism, mutualism) or negative (competition, amensalism) interactions. These interpretations should be confirmed in other independent experiments. The study of bacterial interactions during storage and the identification of the key bacteria that cause food spoilage and foodborne illness will contribute to the development and implementation of effective control strategies to ensure food safety.

## Figures and Tables

**Figure 1 foods-11-03733-f001:**
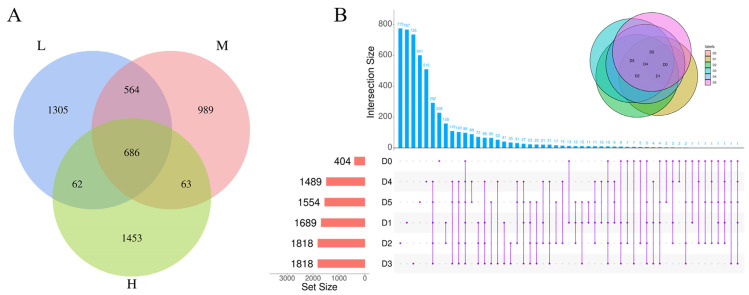
Venn diagram (**A**) and UpSet plot (**B**) based on OTUs of RTE chicken meat bacteria. L, M and H represent samples collected at 4, 8 and 22 °C, respectively. D0–D5 represent samples collected from Day 0 to Day 5.

**Figure 2 foods-11-03733-f002:**
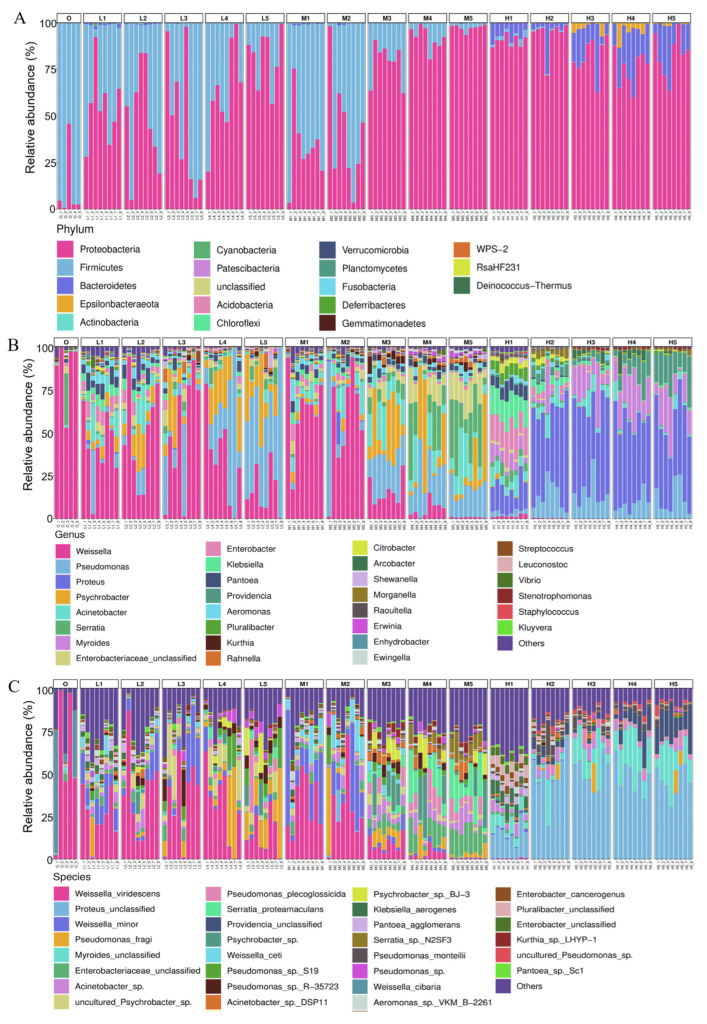
Relative abundance (%) of bacterial taxa based on 16S rDNA sequencing. (**A**) Phylum, (**B**) genus and (**C**) species levels in the RTE chicken meat samples during storage. The multi-colored stack bar graphs display the relative abundances of bacteria in each sample.

**Figure 3 foods-11-03733-f003:**
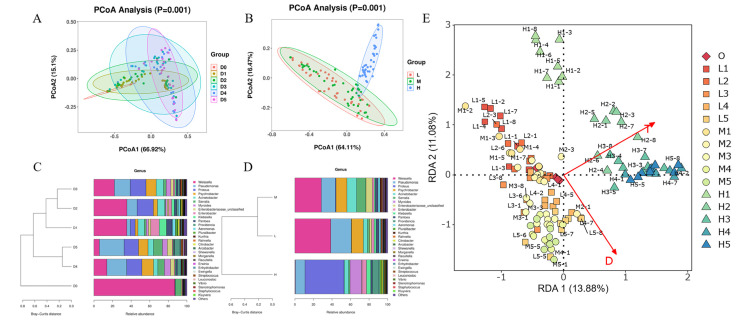
Principal component analysis based on (**A**) storage time and (**B**) temperature. Cluster analysis in the RTE chicken meat samples based on (**C**) storage time and (**D**) temperature at the genus level. (**E**) Redundancy analysis (RDA) of correlations between environmental factors and bacterial community composition.

**Figure 4 foods-11-03733-f004:**
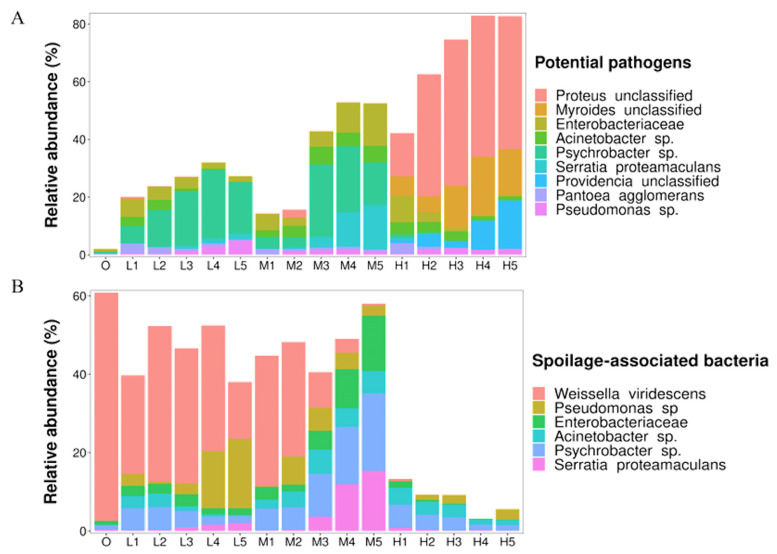
Relative abundance of potentially pathogenic (**A**) and spoilage bacteria (**B**).

**Figure 5 foods-11-03733-f005:**
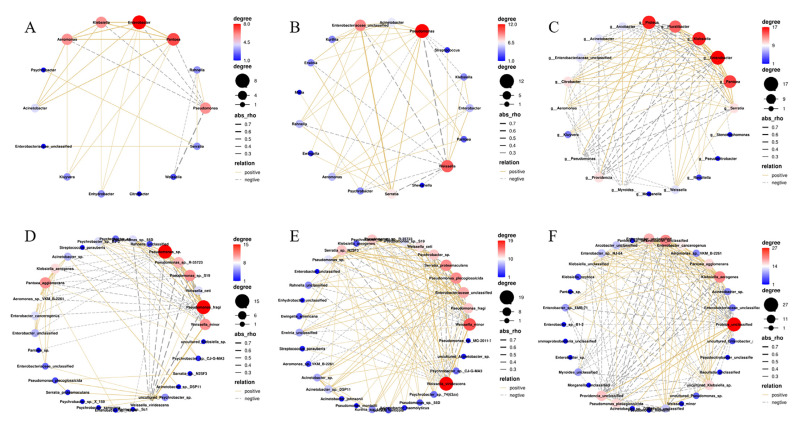
Microbial association networks calculated using SparCC, and the relative abundance data for (**A**–**C**) genera and (**D**–**F**) species for storage temperatures of 4, 8 and 22 °C, respectively.

**Table 1 foods-11-03733-t001:** Diversity indices of RTE meat samples ^†^.

Sample	Observed OTU	Shannon	Simpson	Chao1	Good’s Coverage
O	116 ± 61	2.12 ± 1.45	0.49 ± 0.32	117.47 ± 61.65	0.9998
L1	365 ± 62	5.41 ± 0.76	0.90 ± 0.07	373.24 ± 60.49	0.9996
L2	334 ± 95	4.62 ± 1.75	0.78 ± 0.24	337.67 ± 94.39	0.9997
L3	248 ± 68	3.99 ± 1.35	0.78 ± 0.13	250.68 ± 68.57	0.9998
L4	208 ± 66	3.94 ± 0.77	0.80 ± 0.10	209.78 ± 65.85	0.9999
L5	179 ± 51	4.37 ± 0.78	0.88 ± 0.06	180.21 ± 50.95	0.9999
M1	292 ± 85	4.32 ± 1.22	0.79 ± 0.12	296.57 ± 84.04	0.9997
M2	242 ± 80	4.23 ± 1.19	0.81 ± 0.15	245.80 ± 78.77	0.9998
M3	253 ± 30	5.54 ± 0.33	0.95 ± 0.02	254.55 ± 30.50	0.9999
M4	225 ± 29	5.04 ± 0.61	0.91 ± 0.05	227.37 ± 29.61	0.9999
M5	237 ± 29	5.18 ± 0.57	0.92 ± 0.03	238.79 ± 29.06	0.9999
H1	289 ± 43	6.44 ± 0.14	0.97 ± 0.01	291.01 ± 43.59	0.9999
H2	235 ± 39	4.92 ± 0.36	0.90 ± 0.02	236.52 ± 38.55	0.9999
H3	228 ± 35	4.68 ± 0.46	0.90 ± 0.02	228.64 ± 34.72	0.9999
H4	177 ± 26	4.22 ± 0.23	0.89 ± 0.03	177.89 ± 25.79	0.9999
H5	189 ± 28	4.23 ± 0.37	0.89 ± 0.02	189.92 ± 27.73	0.9999

^†^ Data are expressed as the mean ± standard deviation of the same samples. Samples collected from 4, 8 and 22 °C are named L1 to L5, M1 to M5 and H1 to H5, respectively, and the numbers (1 to 5) represent the storage time (d).

**Table 2 foods-11-03733-t002:** Cardinal symptoms by potential pathogens and spoilage-associated bacteria.

	Species	CardinalSymptom/CorruptionPhenomenon	Reference
Potential pathogens	*Proteus unclassified*	Urinary tract infections, gastroenteritis and wound infections	[31,32]
	*Myroides unclassified*	Urinary tract infections, skin and soft tissue infections, bacteremia, pneumonia and intra-abdominal infections	[33]
	*Enterobacteriaceae*	Urinary tract infections, septicemia, pneumonia, peritonitis, meningitis and intra-abdominal infections	[34,35]
	*Acinetobacter* sp.	Urinary tract infections, skin infections, bacteremia, pneumonia, meningitis and endocarditis	[36,37]
	*Psychrobacter* sp.	Conjunctivitis, endocarditis, peritonitis, bacteremia, infant meningitis, arthritis and surgical wound infections	[38]
	*Serratia proteamaculans*	Pneumonia	[39]
	*Providencia unclassified*	Diarrhea	[40]
	*Pantoea agglomerans*	Septicemia	[41]
	*Pseudomonas* sp.	Bacteraemia	[42]
Spoilage-associated bacteria	*Weissella viridescens*	Produces peroxide which reacts with meat pigment and forms a green-colored	[43,44]
	*Pseudomonas fragi*	slime formation	[45]
	*Enterobacteriaceae*	Forms biofilm and produces gas; putrescine and cadaverine	[46]
	*Acinetobacter* sp.	Produces some volatile spoilage compounds and sulfurous, rancid and fishy off-odors	[47]
	*Psychrobacter* sp.	Produces some volatile spoilage compounds and musty off-odors	[38,48]
	*Serratia proteamaculans*	Produces trimethylamine, putrescine, cadaverine and off-odors	[49]

**Table 3 foods-11-03733-t003:** Microbial interactions detected by SparCC in the bacterial communities of chicken meat samples stored at 4, 8 and 22 °C.

Temperature (°C)	Interaction	Pearson Correlation ^†^	Relation
4	*Enterobacter* ↔ *Pantoea*	0.583	Copresence
	*Klebsiella* ↔ *Pantoea*	0.524	Copresence
	*Aeromonas* ↔ *Pantoea*	0.515	Copresence
	*Enterobacter* ↔ *Klebsiella*	0.498	Copresence
	*Aeromonas* ↔ *Enterobacter*	0.485	Copresence
	*Aeromonas* ↔ *Klebsiella*	0.464	Copresence
	*Acinetobacter* ↔ *Pantoea*	0.420	Copresence
	*Pseudomonas* ↔ *Weissella*	−0.702	Mutual exclusion
8	*Enterobacteriaceae_unclassified* ↔ *Serratia*	0.740	Copresence
	*Pseudomonas* ↔ *Serratia*	0.469	Copresence
	*Enterobacteriaceae_unclassified* ↔ *Weissella*	−0.510	Mutual exclusion
	*Serratia* ↔ *Weissella*	−0.563	Mutual exclusion
	*Pseudomonas* ↔ *Weissella*	−0.677	Mutual exclusion
22	*Enterobacter* ↔ *Klebsiella*	0.664	Copresence
	*Klebsiella* ↔ *Pluralibacter*	0.556	Copresence
	*Enterobacter* ↔ *Pantoea*	0.548	Copresence
	*Klebsiella* ↔ *Pantoea*	0.539	Copresence
	*Enterobacter* ↔ *Pluralibacter*	0.538	Copresence
	*Enterobacter* ↔ *Serratia*	0.479	Copresence
	*Proteus* ↔ *Providencia*	0.462	Copresence
	*Citrobacter* ↔ *Klebsiella*	0.457	Copresence
	*Klebsiella* ↔ *Serratia*	0.455	Copresence
	*Pantoea* ↔ *Pluralibacter*	0.439	Copresence
	*Pantoea* ↔ *Serratia*	0.438	Copresence
	*Proteus* ↔ *Pseudomonas*	0.421	Copresence
	*Citrobacter* ↔ *Enterobacter*	0.403	Copresence
	*Pantoea* ↔ *Pseudomonas*	−0.425	Mutual exclusion
	*Klebsiella* ↔ *Pseudomonas*	−0.444	Mutual exclusion
	*Proteus* ↔ *Serratia*	−0.445	Mutual exclusion
	*Enterobacter* ↔ *Pseudomonas*	−0.445	Mutual exclusion
	*Pantoea* ↔ *Providencia*	−0.467	Mutual exclusion
	*Klebsiella* ↔ *Providencia*	−0.476	Mutual exclusion
	*Pluralibacter* ↔ *Proteus*	−0.489	Mutual exclusion
	*Acinetobacter* ↔ *Providencia*	−0.502	Mutual exclusion
	*Enterobacter* ↔ *Providencia*	−0.503	Mutual exclusion
	*Pantoea* ↔ *Proteus*	−0.530	Mutual exclusion
	*Klebsiella* ↔ *Proteus*	−0.572	Mutual exclusion
	*Enterobacter* ↔ *Proteus*	−0.578	Mutual exclusion

^†^ The Pearson correlation threshold values in the three constructed networks were >0.4 or <−0.4.

## Data Availability

The 16S rDNA sequencing data were submitted to the NCBI Sequence Read Archive (SRA) database, accession no. PRJNA744008.

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
