# Peer review of "Dynamic Changes of Bacterial Communities and Microbial Association Networks in Ready-to-Eat Chicken Meat during Storage"

_foods, 2022, doi:10.3390/foods11223733_

Round 1

Reviewer 1 Report

This Manuscript suffers some major fails in the disposition, terminology, writing style, readability and fluency. Nonetheless, the manuscript could be much improved and could be a useful contribution to our knowledge on the subject and these minor revisions should be addressed:

1- English should improve by a native person.  The paper suffers from a poor English structure throughout and cannot be published or reviewed properly in the current format. The manuscript requires a thorough proofread by a native person whose first language is English. The instances of the problem are numerous and this reviewer cannot individually mention them. It is the responsibility of the author(s) to present their work in an acceptable format. Unless the paper is in a reasonable format, it should not have been submitted.

2- The novelty of the study needs to be highlighted compare to other similar studies or consider to explicitly mention what is gap knowledge and/or what was lacking in the indicated studies.

3- Discussion is weak. The discussion needs enhancement with real explanations not only agreements and disagreements. Authors should improve it by the demonstration of causes of obtained results. Instead of just justifying results, results should be interpreted, explained to appropriately elaborate inferences. Discussion seems to be poor, didn't give good explanations of the results obtained. I think that it must be really improved. Where possible please discuss potential mechanisms behind your observations. You should also expand the links with prior publications in the area, but try to be careful to not over-reach. For the latter, you should highlight potential areas of future study.

4- The scientific background of the topic is poor. In "Introduction" and "Discussion", the authors should cite recent references between 2012-2022 from JCR journals (with impact factor) about recent achievements on the subject. For example, authors should cite to:

Ahir V.B., Singh K.M., Tripathi A.K., Mathakiya R.A., Jakhesara S.J., Koringa P.G., Rank D.N., Jhala M.K. and Joshi C.G. (2012). Study of bacterial diversity in poultry gut using denaturing gradient gel electrophoresis. Iranian J. Appl. Anim. Sci. 2(3), 227-232.

Mirhosseini, S. Z., Seidavi, A. R., Shivazad, M., Chamani, M., Sadeghi, A. A. and Pourseify, R. 2010. Detection of Clostridium spp. and its relation to different ages and gastrointestinal segments as measured by molecular analysis of 16S rRNA genes. Brazilian Archives of Biology and Technology (BABT). 53(1): 69-76

5- A detailed "Conclusion" should be provided to state the final result that the authors have reached. Please note you only need to place your conclusion and not keep putting results, because these have already been presented in the manuscript.

6- Author(s) should re-format the references based on journal format. See the instructions for authors.

7- The numbers and decimals in Tables should be follow the rule of: xxxx, xxx, xx.x, x.xx, 0.xxx and 0.0xxx

Reviewer 2 Report

1. "Abstract", Line 15: Instead of "... its food safety due to bacterial contamination ..." should be re-written as "... its lack of food safety due to ...". 

2. "Abstract", Line 31: "... to reduce foodborne illness outbreaks." should be improved as "... to reduce outbreaks of foodborne illnesses."

3. "Introduction" Line 39-40: Popularity of RTE chicken is also due to consumers' convenience. The authors should modify the sentence accordingly. 

4. Figure 3: The figure legend has mentioned the parts A, B, D, E, F, whereas the figure itself has the parts labelled as A, B, C, D and E. 

5. "Results", sub-section 3.4: Was there any specific analyses of virulence conducted to conclude "potential" pathogenicity of certain genera, separately labeled as such in the Figure 4A? 

Reviewer 3 Report

The manuscript entitled: Dynamic changes of bacterial communities and microbial association networks in ready-to-eat chicken meat during storage by Qiu et al. investigated the impact of storage time and temperature on bacterial compositions and microbial association networks in RTE chicken by using 16S rDNA high-throughput sequencing.

 This manuscript reports results from a fairly well-conducted study. The authors do a good job of introducing the background and rationale for their study and the objectives are clearly defined.

The materials and methods are clearly and concisely described and appropriate to achieve stated objectives. Moreover, The authors adequately assessed the results with the statistical method.

The results were well presented and discussed and the conclusions reached are supported by the evidence they obtained. Overall, the manuscript is interesting and worthwhile. However,  the manuscript needs some minor revisions.  Please find below my comments:

Line 182: Detailed information should be given at the bottom of figure 1 to explain the image's content. The explanation of the codes (L, M, and H) should be rewritten at the figure bottom.

Line 250-262: Microorganism names (E. coli, Acinetobacter, Psychrobacter, Serratia proteamaculans etc.) should be italic.

Line 252, Line 350: Serratia. proteamaculans should be Serratia proteamaculans

Line 257: “at 22°C.   Enterobacteriaceae” should be “at 22°C. Enterobacteriaceae”

Line 255: Authors should give their results in this sentence and no reference is needed.

Line 255, 280-282: The sentences, which are given references, are more appropriate for the discussion section. Likewise, Table 2 should be transferred to the discussion section.

Also, bacteria species should be written in their full name (Listeria monocytogenes) where they are first mentioned, and then their short form (L. monocytogenes).
